# Research on Supply Chain Decisions for Production Waste Recovery and Reuse Based on a Recycler Focus

**Xingyao Liu** [1], **Kai Wang** [1] **and Hui Lu** [2,3,*]

1. School of Business, Jiangsu Ocean University, Lianyungang 222000, China
2. School of Management, Fudan University, Shanghai 200433, China
3. Management School, Shanghai Jianqiao University, Shanghai 201306, China
* Correspondence: luhui8778@yahoo.com

**Abstract:** Production waste recovery has economic and environmental benefits but carbon quotas limit it. To avoid future high-cost recovery technologies, we present an example of Starkelberg model between a recycler and a manufacturer, focusing on recycling exhaust gases containing metal elements from production waste. From the perspective of a recycler, this paper outlines the process of waste recycling, processing, and sales, highlighting how the proportion of recovered waste gas directly or indirectly affects sales volume. The study shows that the impact of different factors on sales volume is bifurcated, with transaction factors affecting both ordinary and new products negatively, while technical aspects positively impact new products. Surprisingly, manufacturers and recyclers benefit, even if the reasons for selling ordinary and new products are different. In the end, the products of ordinary and new in the market are mutual constraints and mutual influence.

**Keywords:** production waste recovery; recovered exhaust gas; remanufactured products; recycle profits

## 1. Introduction

The drive towards low-carbon solutions has heightened the focus on carbon constraints, including carbon quota and carbon trading. The European Union, being at the forefront of this movement, has updated its Waste Framework Directive to allow companies to recycle more in-demand recycled textiles used in the textile and garment industry [1]. Simultaneously, the Environmental Protection Bureau in China has found that the waste gases discharged from the production waste are also harmful to the environment [2]. It can thus be seen that production waste recovery can have economic and environmental benefits.

Carbon trading is also involved in the utilization of production waste. To promote sustainable practices, many countries have implemented laws and regulations related to carbon trading [3]. However, these laws and regulations can be seen as a temporary solution that only treats the symptoms, not the root cause [4]. The reason is that carbon constraints constrain market behavior and have adverse effects during production waste.

On the one hand, carbon constraints constrain the market behavior of enterprises' voluntary trading. They noted that the market behavior is not spontaneous but due to the policy supervision penalty. For example, according to the Ministry of Ecology and Environment, the total value of carbon emission quotas reached 179 million tons, with a policy supervision penalty of 7.661 billion yuan and a 99.5% completion rate of performance [5]. On the other hand, the carbon quota for production waste has not been fully achieved. Germany, for instance, the world leader in recycling, only recycles 16% of composite products [6]. The complex technology, separation, and processing required to recycle production waste often result in only a small part of it being reused.

It is an excellent opportunity to propose that enterprises themselves deal with production waste recovery and reuse through another market mechanism. Suzhou Jinhong Gas Co., Ltd., a leader in China's solid waste industry, is the second largest after the new

energy industry [7] and has become one of the few monopolies in China that deals with production waste. This company has the advantage of being a one-stop shop for recovery, production, and purification, making it a fully integrated gas provider. However, this is a challenging task for most ordinary enterprises, as it requires a significant investment and the profits are limited [8]. Even Jinhong regards production waste recovery and reuse as a unique business rather than its main business [9].

In addition, such recovery is mostly energy recovery after incineration rather than reusing production waste. Most of the recovery is incinerated or landfilled in the United States and Britain, i.e., Lidl in Germany and Mura in Britain [10]. Lidl is one of Germany's most considerable waste recycling retail giants by controlling the overall energy recycling cycle through the production and sales of goods [11]. Mura is finding a third-party partner to recycle the plastic that has been incinerated and buried. Both sides have fulfilled their commitment to environmental sustainability so far [12]. It can be seen that most situations cannot recycle production waste, and the reuse of production waste is also rare.

Most research is related to the process of end-of-life product recovery, treatment, and reuse, which reference consumers' low-carbon preference and the recycling conditions. It should be noted that consumer preference should consider the price difference between new and remanufactured products [13]. Sometimes the price of low-carbon new products and ordinary waste may be the opposite [14]. Additionally, product recovery value and price conditions can affect the decision-maker's choice of social challenges [15]. Therefore, by improving the recovery conditions, the rate of production waste recovery can be increased, bringing economic and environmental and social benefits to the entire system.

Above all, this paper considers that enterprises deal with production waste through a new market mechanism. The relevant regulations on carbon constraint now specifically require enterprises for environmental–social–economic sustainable development to pay attention to such issues again [16]. Therefore, production waste recovery and reuse should carefully consider the recyclers' behavior in the market mechanism when analyzing the carbon quota situation. The reason is that the manufacturer can supply the production waste and process ordinary products.

Moreover, recyclers can sell new products to the market, competing with the ordinary products sold by the manufacturer. The recycler acts as a dealer simultaneously and has a transaction relationship with the producer. Other vital points we explore in this paper are the following:

(1) How can the manufacturers guarantee a profit increase when supplying the exhaust gas to recyclers?
(2) How do recyclers take the best decision in the process of production waste recovery and reuse with or without a carbon quota?
(3) Should the government participate in production waste recovery and the reuse process to make the scheme more economical and effective?
(4) What is the market impact of different factors on the decision-making of ordinary and new products?

It should be noted that the recycler is usually a third party and is affected by consumers' preferences, competition, cost, and other factors that influence consumers' sales decisions and profits [17]. Therefore, recyclers must consider a different price strategy when studying the coordination mechanism to achieve their best profit [18]. For example, Suzhou Moore Gas Co., Ltd. recovers nitrogen from various trades, including electronics, the chemical industry, optical fiber, and metallurgy, to achieve the best profit [19]. Because manufacturers choose different channels, their wholesale prices and retailers' marketing effects will be distinguished [20]. Therefore, this paper uses the recycler as the third party when building its model.

The result was beyond our expectations when the proportion of waste gas recovery was proposed for recovering gas containing metal elements from the production waste process. This study introduces a novel approach to connect the traditional supply chain to the gas supply chain, providing valuable insights into the management field and helping

enterprises fully grasp the benefits of product quality assurance. As a third party in the waste recycling and reuse process, the recycler can enhance the utilization efficiency of products and achieve the sustainable goals of the Green supply chain. More attention here is paid to the influence of specific factors in the research process than before when taking the proportion of recovered exhaust gas as a new entry point of technological innovation.

The rest of this paper is organized as follows: Section 2 reviews the relevant literature. Section 3 describes model problems and fundamental assumptions. Section 4 analyzes the optimal results without considering the carbon quota and then considering the carbon quota. Section 5 offers a comparative analysis of the above two situations. Section 6 is an example analysis. Section 7 summarizes the paper and its findings and expounds on the significance of this study for the future development of production waste recovery and its reuse.

## 2. Literature Review

### 2.1. Waste Recycling and Reuse Analysis

This paper focuses on production waste recovery and reuse. While most studies on recovery, treatment, and reuse have concentrated on ordinary or end-of-life products. [21]. It has also been discovered that there is no significant interaction between the different waste recovery rates, renewable energy, and carbon dioxide emissions in terms of product recovery rate [22], which is related to advertising efficiency and product order quantity [23].

When there is no third-party involved, producer-led independent recycling channels are the most efficiency for channel selection [24]. These channels can sometimes maximize the total profit of the supply chain through government subsidies [25]. At that time, manufacturers and recyclers will conduct price competitions and cooperation [26]. In contrast, when considering the third party, the best choice is to use the recycling platform for old goods recycling [27]. Specifically, such competition between third-party recyclers can reduce product prices and improve the recycling efficiency of waste products but also play a positive role in promoting income level [28].

Most scholars have adopted the empirical research method to consider production waste recovery, treatment, and reuse. However, the effectiveness and efficiency of municipal production waste recycling are unsatisfactory due to poor waste segregation at the generation source [29]. Using field observations, semi-structured interviews, and secondary data sources, the structural model method can better analyze the interaction in the end [30].

However, there are few theoretical models on the complete production waste recycling and reuse process. Even if there were, it would refer to unilateral recycling of production waste [31]. Moreover, theoretical models convert discarded waste into energy for utilization [32]. Analyzing the public's perception of risk is thus the best way to deal with waste-to-energy [33]. Some documents, when considering third parties, pay attention to everyone, when sending a revised version of MS to the reviewers, but we only need to highlight the modified content, without having track changes on, and without emphasizing any deleted content, and so function as a recycling platform in this way [34]. Therefore, the case suggests that production waste recycling solves the pricing and profit problem by analyzing the relationship between the recycling rate and consumers' recycling willingness [35].

### 2.2. An Analysis of Remanufacturing

This analysis aligns with the public's idea that remanufactured products are not as good as new ones. That recognition will lead many people to believe that remanufactured products should be "the cheaper, the better" [36]. The challenge remanufacturers face is that it can be difficult to accurately determine the condition of used products before disassembly and inspection [37]. However, research has shown that while the optimal sales price of new and remanufactured products increases, the recovery price and social welfare decrease [38]. Distribution channel choice [39] thus should be used as a potential factor to foster more remanufactured products [40] and their sales [41].

It should be noted that while manufacturers can benefit from adopting emission reduction technology, recycling, and a remanufacturing approach, they cannot guarantee the minimum total carbon emissions in the supply chain [42]. In addition, remanufacturing should consider the impact of policy because the distribution of government subsidies has an incentive effect on both manufacturers and consumers [43]. However, remanufacturers must also meet other requirements, such as obtaining patent authorization and addressing equity concerns, to turn a profit [44]. Remanufacturing balances economic interests and environmental impact more effectively than tax and carbon reduction policies [45]. Therefore, many scholars have proposed that more insight is required into consumer expectations and willingness [46]. For example, the research should focus on differential pricing and the profit distribution of new and ordinary products [47].

*2.3. Analysis of the Carbon Quota*

By combining transaction prices and incorporating multiple entities, the allocation of carbon quota can have varying effects. Therefore, in any allocation and use of carbon quota, a government enterprise consumer tripartite carbon quota allocation can be established [48]. It can also improve the participation enthusiasm of consumers and enterprises based on maintaining the best social welfare. Thus, both meet fairness and efficiency requirements and lower emission reduction costs [49].

Although the carbon quota policy can improve the sustainable level of inventory and transportation, it has no impact on the service level [50]. Therefore, when considering the carbon quota and its transaction price, enterprises need to pay more attention to environmental and consumers' Green preferences, such as marketing strategies related to advertising [51] and sales [52], as well as Green goodwill [53]. Indeed, given differential pricing and unified pricing under a carbon quota, enterprise profits will be affected by differences in competition and cooperation [54].

*2.4. Waste Audit in the Supply Chain*

The field of production waste recovery and reuse holds immense potential and value, as it remains a relatively untapped area of research. Despite this, the existing studies have primarily focused on solid waste and have not yet fully explored the implications of a carbon quota in the waste recovery and treatment process.

Supply chain audit is related to cost and inter-subject competition [55]. On the one hand, partners input costs to operate a prediction market for sharing their forecasts and resolving their differences through costly actions [56]. On the other hand, competition among different entities can harm audit quality but can favor cost pricing [57]. Even waste audits can be used as an environmental management tool to help regulators grasp information on waste management and guide enterprises to carry out waste risk management [58]. Surprisingly, when audits are tied to recycling, they can drive green and sustainable development [59].

In conclusion, there is much room for exploration in the theoretical production waste recovery and reuse model. The research on recycling and reusing production waste also focuses on an empirical research method. Still, the review of the related literature suggests that the existing studies on waste recovery and treatment have mainly focused on solid waste. Besides, the market mechanism of production waste can be attractive when considering the carbon quota because most of the research on the carbon quota has nothing to do with the recycling and reuse of production waste. This research can examine the influence of different factors, such as cost and competition, on the carbon quota. In the next section, we present a description of our modeling framework.

## 3. Model Preparation

*3.1. Problem Description*

In this paper, a supply chain model consisting of a manufacturer and a recycler is considered from the recycler's perspective, taking waste gas recovery as an example. In

detail, there are two sales channels: (1) the manufacturer produces ordinary products and deals with production waste; (2) the recycler then recycles the production waste of ordinary products and makes new products for sale to the market. It's worth mentioning that recycling refers to the exhaust gas containing metal elements during waste production. On the one hand, the recycler can recover the waste gas from the production waste collected from the waste discharged from the manufacturer's ordinary products. On the other hand, recyclers convert the recovered gas into new products and sell them to the market. The model's overview is illustrated in Figure 1.

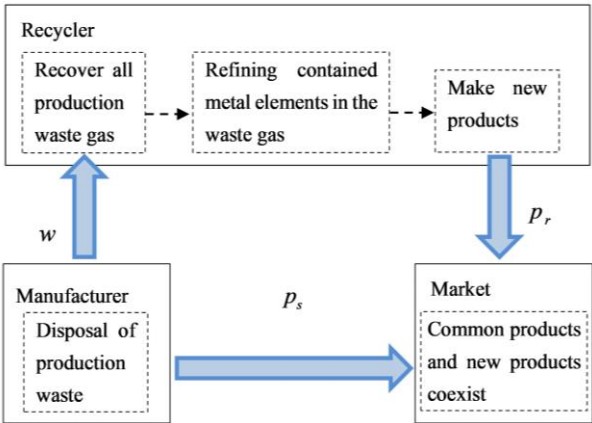

**Figure 1.** Model diagram for waste recycling.

Some similarities and differences should be noted as well. Like [20,47], manufacturers choose different channels to distinguish between new and ordinary products' prices. There are ordinary products that have not been recycled and new products that have been recycled and reused in the market. Under the Stackelberg game, the manufacturer is the leader, and the recycler is the follower. The difference is that the game is played between the producer and the recycler, resulting in different prices, costs, and sales volumes. It is also important to note the wholesale price.

Combined with [49,54], we must consider the following problems. First, we must consider market trading risks to guarantee the manufacturers' profit. Second, we can analyze whether recyclers need to consider carbon emissions when making optimal decisions. Next, we can explore whether this model result is affected by different factors. Last but not least, we need to consider which scheme is more economical in the situation of competition and cooperation for the government. Therefore, enterprises need to consider cost, sales volume, profit, recovery ratio, and other factors in detail.

*3.2. Basic Assumptions*

The fundamental assumption is that the recycler recovers the waste gas containing metal elements to make new products. In detail, if a unit of ordinary product can emit $m$ tons of waste gas, the total emission of exhaust gas is $m \cdot q_s$ when the carbon quota is not considered and are $\beta \cdot m \cdot q_s$ when the carbon quota is considered. Taking the former equation as an example, if the proportion of waste gas recovered by the recycler is $\tau(0 < \tau < 1)$, then the amount of waste gas recovered is $\tau \cdot m \cdot q_s$; if the gas concentration ratio is $\delta(0 < \delta < 1)$, then the amount of gas containing metal elements is $\delta \cdot \tau \cdot m \cdot q_s$; if the ratio between the gas containing metal elements and the new product is $\sigma$, then the number of new products sold by the recycler is $q_r = \frac{\delta \cdot \tau \cdot m}{\sigma} q_s$ (Table 1). Therefore, it can be inferred that:

(1)    the profit function for the recycler is $\pi_r = (p_r - c_r) \cdot \frac{\delta \cdot \tau \cdot m}{\sigma} q_s - c_t \cdot \delta \cdot m \cdot \tau \cdot q_s - w \cdot \tau \cdot m \cdot q_s$,

(2)    the profit function of the manufacturer is $\pi_m = w \cdot \tau \cdot m \cdot q_s + (p_s - c_s) \cdot q_s$,

(3)    the demand function of new products is $p_r = \theta \cdot [Q - q_s(\frac{\delta \cdot \tau \cdot m}{\sigma} + 1)]$,

(4)    the demand function of ordinary products is $p_s = Q - q_s(\frac{\theta \cdot \delta \cdot \tau \cdot m}{\sigma} + 1)$.

**Table 1.** Parameters and Symbols of Production Waste Process.

| Symbol | Illustration |
|:---:|:---:|
| $Q$ | Potential market demand |
| $m$ | The quantity of exhaust gas that can be generated per unit of ordinary products ($m > 0$) |
| $\beta$ | The proportion of allowable emissions affected by the carbon quota ($\beta \in (0,1]$) |
| $w$ | The production cost of wholesale waste from the manufacturer to the recycler |
| $\tau$ | The proportion of recovered exhaust gas |
| $\delta$ | Gas concentration ratio |
| $\sigma$ | Gas conversion ratio |
| $c_s$ | The production cost for the manufacturer selling ordinary products to the market ($c_s > c_r/c_t$) |
| $c_t$ | The production cost for recycler refining gas containing metal elements |
| $c_r$ | The production cost of new products made by recyclers |
| $p_s$ | The unit price for ordinary products the manufacturer sells to the market |
| $p_r$ | The unit price at which the recycler sells new products to the market ($p_r > p_s > w$) |
| $q_s$ | The number of ordinary products sold by the manufacturer |
| $q_r$ | The number of new products sold by the recycler ($q_r \geq q_s \geq 0$) |

When considering the carbon quota situation and so on, this study does not elaborate. Other basic assumptions for the model notions are as follows:

(1)    All new products will have the same quality and can be remanufactured and sold.

(2)    To simplify the model expression, let $Q = 1$, $\frac{\delta}{\sigma} = 1$, $m = 1$. When the carbon quota is not considered, $q_r = m \cdot q_s$ when the carbon quota is considered,

$$q_r = \beta \cdot \tau \cdot q_s \tag{1}$$

(3)    Consumers' willingness to buy ordinary and new products may differ. In other words, consumers' acceptance of ordinary and new products has a consumer psychological preference $\theta$, and $0 < \theta < 1$. Currently, the demand function is

$$p_r = \theta \cdot [1 - q_s(\tau + 1)] \tag{2}$$

$$p_s = 1 - q_s(\theta \cdot \tau + 1) \tag{3}$$

## 4. Remanufacturing Model Results

### 4.1. Without Considering the Carbon Quota

It should be noted that the influencing factors in this paper can be roughly divided into two categories in terms of production waste recycling, treatment, and reuse: technical factors and transaction factors. In particular, the technical factors are mainly the carbon quota ratio, gas concentration (conversion) ratio, and recycled waste gas ratio that need attention for the process technology. However, transaction factors are price and cost in sales businesses.

In addition, the waste gas has gone through recovery, refining, and sales in a single cycle. When the price of wholesale gas is $w$, the sales price of the products sold by the manufacturer to the market and the price of the recycler are $p_s$ and $p_r$, respectively. If

necessary, a distinction needs to be made between general and optimal cases of factors, such as sales volume and price.

Based on the sales behaviors of a manufacturer and a recycler, the formulas for the manufacturer to sell ordinary products and the recycler to sell new products are as follows:

The profit function for the manufacturer is:

$$\pi_m = w \cdot \tau \cdot q_s + (p_s - c_s)q_s \tag{4}$$

The profit function for the recycler is:

$$\pi_{r1} = (p_r - c_r - \delta \cdot c_r - w) \cdot \tau \cdot q_s \tag{5}$$

**Proposition 1.** *The cost of gas-containing metal elements in the recovery, treatment, and reuse of production waste has a negative impact on producers but a positive impact on recyclers.*

See Appendix A for the proposition text and inference process in this paper.

The cost of containing metal elements is the initial price of refining metal-containing gases. Its impact on different channels reflects the changes in manufacturers and recyclers in some respects. Specifically, the adverse effects of gas-containing metal elements on ordinary products are reflected in the price of wholesale gas and the sales volume of ordinary products. In contrast, the positive impact on new products is reflected in the sales volume of those new products in Corollary 1 and Corollary 2 below.

**Corollary 1.** *Without considering the carbon quota, the optimal price of a producer's wholesale gas decreases with an increase in the cost of the recycler's refining of the gas-containing metal elements.*

**Corollary 2.** *Without considering the carbon quota, the optimal sales volume of ordinary products decreases with an increase in the cost of recovering the gas-containing metal elements by the recycler, while the optimal sales volume of new products increases with an increase in the cost for retrieving the gas-containing metal elements.*

The results from Corollarys 1 and 2 reveal that cost is one of the primary transaction factors affecting the optimal sales volume of ordinary products. The increase in the cost of gas-containing metal elements is due to the poor quality control measures for waste recovery products, which increases the scrap cost [60]. However, with the support from the government for the growth of the environmental protection sector, the optimal wholesale gas cost still tends to decrease, even with rising scrap costs [61]. This prompts recyclers to purchase more waste gas at a lower wholesale cost and sell new products with a small profit margin, leading to an increase in the sales volume of new products (as stated in Corollary 5). On the contrary, the optimal sales volume of ordinary products is reduced by the competition with similar products in the market. This reinforces the notion that supply chain waste audits are related to cost, and competition and that waste recovery and reuse are impacted by these factors.

**Proposition 2.** *Ordinary products are negatively affected by transaction factors, such as cost and price, in the process of production waste recovery, treatment, and reuse.*

Here, only the transaction factors affecting the general sales volume and optimal sales volume of ordinary products are explained in Corollary 3. More specifically, the general cost of wholesale gas negatively affects the optimal sales of ordinary products. New products are analyzed in Section 6.2, which mainly discusses technical factors. Furthermore, Corollary 4 summarizes the production cost range of new products, which is generated and extended under the mutual relationship between wholesale pricing and the sales volume of ordinary products.

**Corollary 3.** *Without considering the carbon quota, the optimal sales volume of ordinary products will decrease with an increase in the general cost of wholesale gas.*

Corollary 3 proves that the optimal sales volume of ordinary products and the general cost of wholesale gas change inversely. The details are as follows: when the optimal sales volume is low, the general cost of wholesale harms the optimal sales volume of ordinary products. The reason is that the larger the wholesale price, the more substantial the profit the manufacturer receives from the recycler, and the less the investment in the ordinary products, resulting in a decrease in the optimal sales volume of the ordinary products.

Further, combining Corollary 1 with 2, it is found that the optimal sales volume of ordinary products and the optimal cost of wholesale gas show positive changes. Still, the optimal sales volume of ordinary products and the general cost of wholesale gas show negative changes. It can be seen that cost stability affects the optimal sales volume of ordinary products. In contrast, the wholesale gas cost strongly fluctuates the optimal sales volume of ordinary products.

**Corollary 4.** *When the carbon quota is not considered, and $\theta \in (0,1)$, $c_s \in (0,1)$, $\delta \in (0,1)$);*
(1) *if $\frac{\theta - c_t}{\delta} < 0$ and $\frac{\theta - c_t}{\delta} > 1$, and $2c_s - \delta \cdot c_t + \theta - 2 \in (0,1)$, $c_r \in (0, 2c_s - \delta \cdot c_t + \theta - 2)$;*
(2) *if $0 < 2c_s - \delta \cdot c_t + \theta - 2 < 1 < \frac{\theta - c_t}{\delta}$, $c_r \in (0, 2c_s - \delta \cdot c_t + \theta - 2)$;*
(3) *if $0 < \frac{\theta - c_t}{\delta} < 2c_s - \delta \cdot c_t + \theta - 2 < 1$, $c_r \in \varnothing$;*
(4) *if $0 < 2c_s - \delta \cdot c_t + \theta - 2 < \frac{\theta - c_t}{\delta} < 1$, $c_r \in (2c_s - \delta \cdot c_t + \theta - 2, \frac{\theta - c_t}{\delta})$;*
(5) *if $2c_s - \delta \cdot c_t + \theta - 2 < 0$ and $\frac{\theta - c_t}{\delta} > 1$, $c_r \in (0,1)$.*

**Proposition 3.** *The proportion of waste gas recovered plays an intermediary role in the process of production waste recovery, treatment, and reuse. It has a positive impact on both new products and ordinary products.*

The proportion of recycled waste gas plays a crucial role in the relationship between the cost and sales volume of new and ordinary products. This proportion is a dynamic factor that is affected by and affects other factors in the waste recovery process. In other words, the proportion of waste gas recovered is affected by the cost and the sales volume of new and ordinary products, which is beneficial. The impact on the proportion of waste gas is explained in Corollary 5, while the reason it affects ordinary products and new products is described in Corollary 6. Additionally, the policy and the production cost of new products affect the proportion of recycled waste gas, as illustrated in Corollary 10 and Section 6.2.

**Corollary 5.** *Without considering the carbon quota, the optimal proportion of the recovered exhaust gas increases with the increased cost the recycler pays to extract the gas-containing metal elements.*

**Corollary 6.** *Without considering the carbon quota, the optimal sales volume of ordinary products and the general sales volume of new products will increase with an increased general proportion of recovered exhaust gas.*

Corollary 6 shows that when the optimal proportion is not reached, the proportion of recovered exhaust gas positively impacts the sales volume of both ordinary and new products. Specifically, in terms of technology, the higher the ratio of waste gas recovery, the greater the augment in the amount of waste gas emitted. Because the production of ordinary products emits exhaust gas, the number of ordinary products is increasing. As a result, the increased production of ordinary products leads to a higher sales volume. All in all, different from the wholesale price of gas in Corollary 3, the proportion of recycled waste gas is the second reason that will affect the optimal sales volume of ordinary products. The optimal sales volume of new products is affected not only by the general proportion of waste gas recovered but also by other factors, as detailed in Section 6.2.

**Proposition 4.** *Remanufactured and ordinary products are mutually restricted and occupy the market in the competition.*

The production cost of ordinary products is a key transaction factor that can significantly impact the market. It is not as common for the price of competitive products to decline even if the cost of ordinary products increases. However, the increase in the price of ordinary products causes the pitch in their prices, which is very rare for the price of competitive products to decrease. Proposition 4 indicates that if ordinary products change in a certain respect, new products will also be affected. Moreover, the market will show a declining situation and another rising. The specific description is described in Corollary 7.

**Corollary 7.** *Without considering the carbon quota, the optimal sales volume and optimal price of ordinary products will increase with the production cost of ordinary products. In addition, the optimal sales volume and optimal price of new products will decrease with the increase in the production cost of ordinary products.*

From Corollary 7, the production cost of ordinary products has a positive impact on the optimal price of ordinary products and a negative effect on the optimal price of new products. As with the price transmission and transfer effect of Jinfei Kaida company, the sales volume is the inventory, and the sales price is the sales value [62]. Known sales price = cost × sales volume + other. In other words, sales value = cost × inventory + additional [63]. When the production cost of ordinary products rises, then unsalable ordinary products will increase the inventory, increasing the value of the accumulated sales products. In contrast, when the production cost of ordinary products rises, the inventory of new products decreases due to the substitution of new products, resulting in a decrease in the accumulated value of the new products.

*4.2. Considering the Carbon Quota*

The carbon quota limits the carbon emission amount for factories and enterprises. In essence, the carbon quota limits the total amount of waste gas emitted by manufacturers and thus affects recyclers' recycling, refining, and sales processes. Therefore, the channel for manufacturers to sell ordinary products is unchanged, and the profit function of the recycler is as follows:

$$\pi_{r2} = (p_r - c_r - \delta \cdot c_r - w) \cdot \beta \cdot \tau \cdot q_s \tag{6}$$

**Proposition 5.** *The environmental situation that considers the carbon quota has a significant impact on recyclers, and policy is one of the factors affecting the proportion of waste gas recovered.*

There are some similarities and differences between not considering a carbon quota and considering a carbon quota. Corollary 8 shows that manufacturers have more similar results for changes in different environments. Different parts are primarily from recyclers, which are explored in greater detail in the case study section. It is worth noting that the policy factor is neither a technical factor nor a transaction factor. The influence of policy on the proportion of recovered waste gas is mandatory, as demonstrated in Corollary 10. Additionally, as outlined in Corollary 9, the cost is a limiting factor, but it does not violate conventional wisdom.

**5. Comparison and Analysis**

The results from this study demonstrate the optimal solutions for the two cases, considering and not considering the carbon quota, allowing for decision-making.

*5.1. Comparison and Analysis of Optimal Values*

**Corollary 8.** *The optimal solution satisfies the conditions without considering the carbon quota and the carbon quota (1)* $w_1^* = w_2^*$, $q_{s1}^* = q_{s2}^*$; *(2) Corollary 1, Corollary 2, Corollary 3, Corollary 6, and Corollary 7.*

Corollary 8 demonstrates that when analyzed with the aforementioned findings, the similarities between not considering the carbon quota and the carbon quota mainly relate to the content of trend change, and most similarities are related to the producers. The trend changes in (2) have one side increasing, resulting in the other side also changing. However, as opposed to Corollary 5, it is revealed that the impact of different scenarios has a more pronounced effect on recyclers. In essence, different situations will have a more substantial effect on recyclers than on manufacturers.

**Corollary 9.** *(1) If* $\delta \in (0,1)$, $c_r > c_t$ *and* $c_r/c_t \in (0,1)$, $p_{s1}^* < p_{s2}^*$, $p_{r1}^* < p_{r2}^*$; *(2) If* $\theta \in (0,1)$, $c_r \neq c_t$, *and* $c_r/c_t \in (0,1)$, $\pi_{m2}^* > \pi_{m1}^*$.

Corollary 9 illustrates the point that cost is also a constraint. There may be other costs, but the results can only be suitable under limited conditions. Combined with the above assumptions, we will obtain: $c_s > c_r > c_t$. The relationship between price and profit under certain conditions is also shown here, paving the way for the following analysis.

*5.2. Trend Comparison and Analysis*

**Corollary 10.** *The difference between considering the carbon quota and not considering the carbon quota is that the optimal proportion of the recovered exhaust gas decreases with an increase in the cost for the recycler to extract the gas containing metal elements.*

The results of this model comparison demonstrate the impact of considering or not considering the carbon quota on the proportion of recovered waste gas. According to Corollary 5, the increase in the proportion of recovered waste gas without considering the carbon quota results from the low cost of wholesale gas purchases. On the other hand, reducing the proportion of recovered waste gas when considering the carbon quota is due to the direct limit on the amount of waste gas discharge. Therefore, increasing the cost of refining metal element gas cannot change policy implementation.

**6. Example Analysis**

Some relationships cannot be directly derived by analyzing value from the model process, such as the optimal proportion of recycled exhaust gas, the sales volume, the sales price of ordinary and new products, and the relationship between the profits of recyclers. Therefore, this paper uses the numerical simulation method to explore the above conclusions and obtain more laws for waste recycling. After referring to the internal data of several enterprises to understand the relevant situation, different parameters were set as follows: $c_s = 0.9$, $\theta = 0.5$, $\delta = 0.5$, $c_t = 0.1$. According to the above analysis and without considering the carbon quota, and considering the carbon quota, $c_r \in (0.25, 0.4)$.

*6.1. Comparing and Analyzing the Optimal Proportion of Recovered Waste Gas*

According to the parameters $\tau_1^* = \frac{-11c_r+3.5}{4c_r-1}$, $\tau_2^* = \frac{0.8-2c_r}{c_r-0.25}$, the relationship between the optimal proportion of recovered exhaust gas and without considering the carbon quota and considering the carbon quota in Figure 2. The detailed description of this image is as follows: (1) The optimal proportion of recovered exhaust gas decreases with an increase in the production cost of new products without considering the carbon quota and the carbon quota. (2) The optimal proportion of recovered exhaust gas under consideration of a carbon quota is higher than without any consideration of a carbon quota ($\tau_2^* > \tau_1^*$). Combined

with Corollary 5, the result explains that the cost of gas containing metal elements positively impacts the proportion of recovered waste gas. In contrast, new products' production cost negatively impacts the recovered waste gas ratio.

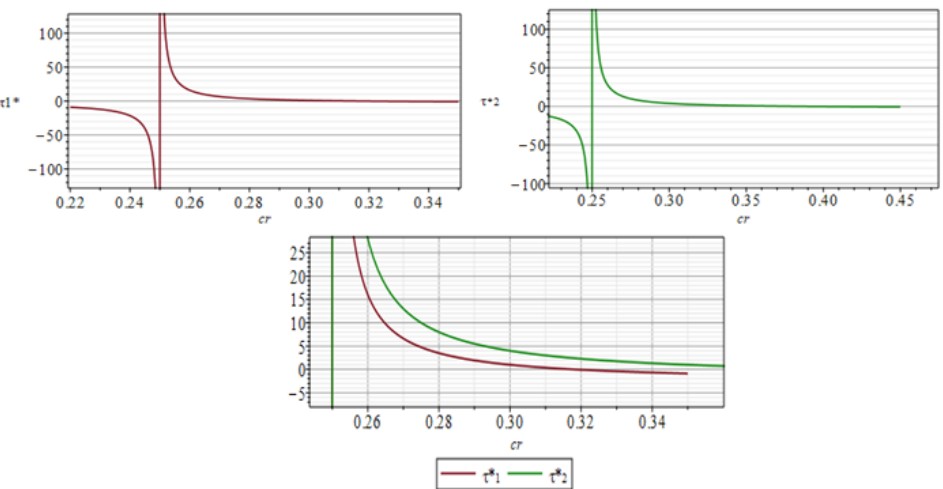

**Figure 2.** The new product production cost affects $\tau_1^*$ and $\tau_2^*$.

### 6.2. The Optimal Sales Volume of New Products under Different Circumstances

According to the parameters $q_{r1}^* = -0.9167c_r + 0.2917$, $q_{r2}^* = -0.6667\beta(c_r - 0.40)$, $q_{r1}^* - q_{r2}^* = -0.9167c_r + 0.2917 + 0.6667\beta(c_r - 0.40)$, the relationship between the optimal sales volume of new products without considering the carbon quota and considering the carbon quota is analyzed in Figure 3. The details are as follows: (1) The optimal sales volume of new products without a carbon quota decreases with the increase of $c_r$, and the optimal sales volume of new products with a carbon quota decreases with the growth of $c_r$, along with the rise of $\beta$. (2) The optimal sales volume of new products without considering a carbon quota is more than that for new products considering a carbon quota. This trend is because $q_{r1}^* - q_{r2}^*$ increases with $c_r \in (0.25, 0.4)$, and $\beta \in (0.25, 0.4)$ shows a trend of at first greater than zero and then less than zero.

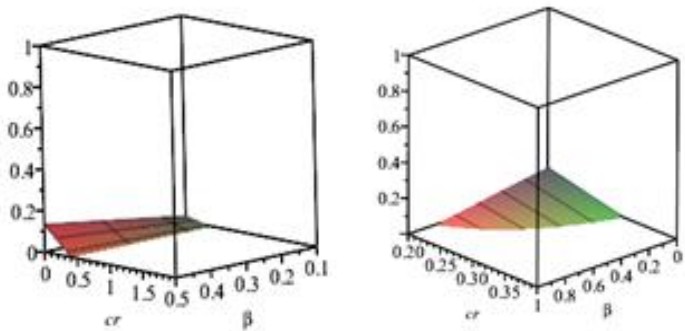

**Figure 3.** The influence of different factors on $q_{r2}^*$ (**left**) and $q_{r1}^* - q_{r2}^*$ (**right**).

Therefore, the cost is one of the main transaction factors that affect the optimal sales volume of ordinary products. Corollary 2 is verified. Combined with Proposition 1, the production cost of new products hurts the optimal sales volume of both ordinary and new products. Moreover, the optimal sales volume of new products is also affected by the carbon quota ratio and the concentration or conversion ratio (which can be obtained by the same process in addition to the cost). The content of the explanation responds to Corollary 6. This positive effect will eventually cause the optimal sales volume of the carbon quota to exceed the optimal sales volume without considering the carbon quota.

It can be seen, therefore, that the Green supply chain will adhere to the carbon quota and gradually improve the quality of raw materials, and finally achieve good sales results. New products are positively affected by technical factors, such as the proportion of recycled waste gas, the carbon quota ratio, and the concentration or conversion ratio, which also proves that the technology mentioned before is one of the factors that cannot be ignored. In addition, transaction factors, such as new product production costs, also hurt new products' sales.

### 6.3. A Comparison and Analysis of Optimal Product Prices

According to the parameters $p_{s1}^* = 0.125c_r + 0.9375$, $p_{s2}^* = 0.9$, $p_{r1}^* = 0.3958 + 0.2917c_r$, $p_{r2}^* = 0.4083 + 0.1667c_r$, the price relationship between ordinary products and new products is analyzed in Figure 4. Synthesized the results of the three maps, without considering the carbon quota and considering the carbon quota, given the increase $c_r$, the optimal sales price of ordinary products is higher than that of new products. That is to say $p_{s1}^* > p_{r1}^*$, $p_{s2}^* > p_{r2}^*$. In addition, since $c_r > c_t$ verifies $p_{s1}^* < p_{s2}^*$, $p_{r1}^* < p_{r2}^*$ for Corollary 7 (1), we get $p_{r1}^* < p_{s1}^* < p_{s2}^*$, $p_{r1}^* < p_{r2}^* < p_{s2}^*$. Compared with $p_{s1}^*$ and $p_{r2}^*$, it can be seen that $p_{s1}^* > p_{r2}^*$ within the range of $c_r$, so $p_{r1}^* < p_{r2}^* < p_{s1}^* < p_{s2}^*$. Therefore, the rising production cost of new products has a positive impact on the optimal price of ordinary products and a negative effect on the optimal price of new products.

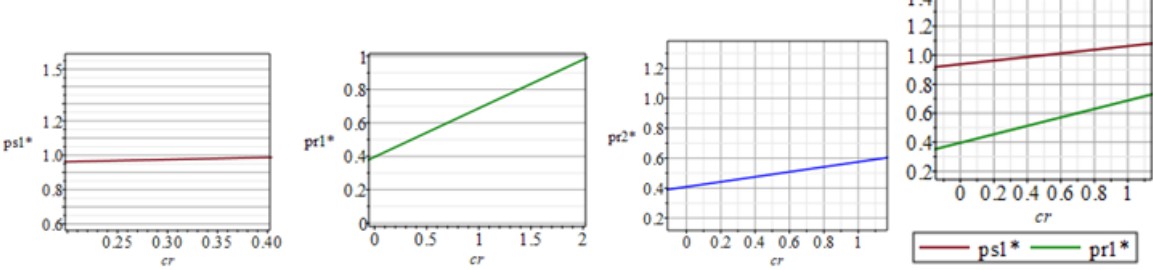

**Figure 4.** The analysis of new product production cost on $p_s$ and $p_r$.

### 6.4. A Comparison and Analysis of Product Optimal Profit

$\pi_{r1}^* = 0.0556(2.75c_r - 0.875)^2$, $\pi_{r2}^* = 0.2222\beta(c_r - 0.40)^2$, $\pi_{m2}^* = 0.05500 - 0.3000c_r + 0.3333(c_r + 0.05)^2$, $\pi_{m1}^* = -1.000(-0.2500c_r + 0.2000)(-0.2917 + 0.9167c_r) + 0.3333(0.1250c_r + 0.0375)(c_r - 0.25)$, the figure thus analyzes the profit relationship between producers and recyclers and is verified by the above parameters that let Corollary 7 (2) get $\pi_{m2}^* > \pi_{m1}^*$. One more note, the color of the picture is to show that it is a 3D picture. Details are as follows:

First, compare the relationship between $\pi_{r2}^* - \pi_{m2}^*$, $\pi_{r1}^* - \pi_{r2}^*$, $\pi_{r1}^* - \pi_{m1}^*$. The details are as follows: (1) Under the consideration of a carbon quota, the overall profit of recyclers is less than that of producers with $c_r$ and $\beta$. As shown panorama of the model in Figure 5a, the color indicates that the darker the range of influence the smaller and the lighter the range of influence the larger. (2) With the increase of $c_r$, the profit of recyclers without considering the carbon quota is less than that of the recyclers without considering the carbon quota first and more remarkable than that of the recyclers without considering the carbon quota. As shown in the detail diagram of the model in Figure 5b, white and color represent the difference, with white representing a difference less than zero and color representing a difference greater than zero. (3) Without considering the carbon quota, the profit of the recycler will be less than that of the producer with the increase $c_r$.

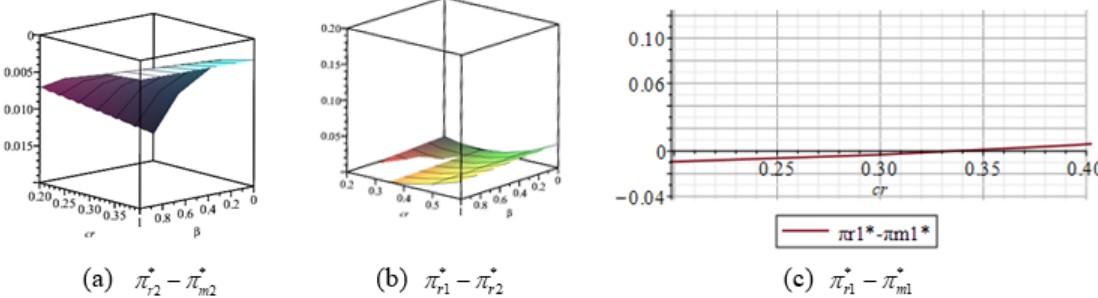

**Figure 5.** The overall profit comparison and analysis chart of $\pi_r$ and $\pi_m$.

It can be seen, therefore, that the optimal profits of recyclers and producers affect each other. Considering that the carbon quota is beneficial to the overall yield of producers, not considering the carbon quota is advantageous to the optimal profit of recyclers. Of them, the rising production cost of new products is conducive to the gain of recyclers without considering the carbon quota; the increase in the proportion of compressed gas is also conducive to considering the profits of recyclers under the carbon quota.

Secondly, based on the above optimal profit, when comparing the manufacturer's internal profit and the recycler's profit, the internal profit of the manufacturer is divided into the profit from wholesale gas and the profit from ordinary products, and the profit of the recycler is then the profit from new products. These results, without considering the carbon quota and considering the carbon quota, satisfy the following requirements: (1) The optimal profit of wholesale gas is more significant than that of ordinary products first and then smaller. Because $\tau_1^* \cdot w_1^* \cdot q_{s1}^* - (p_{s1}^* - c_s)q_{s1}^*$ and $\tau_2^* \cdot w_2^* \cdot q_{s2}^* - (p_{s2}^* - c_s) \cdot q_{s2}^*$ are first greater than zero and then less than zero with the increase of $c_r$. (2) The optimal profit for wholesale gas is less than that for new products first and more significant than for new products. Because $\tau_1^* \cdot w_1^* \cdot q_{s1}^* - \pi_{r1}^*$ with the increase of $c_r$, $\tau_2^* \cdot w_2^* \cdot q_{s2}^* - \pi_{r2}^*$ with the rise of $c_r$ and $\beta$, all are first less than zero and then greater than zero. (3) The optimal profit of ordinary products is less than that of new products. Because $(p_{s1}^* - c_s)q_{s1}^* - \pi_{r1}^*$ and $(p_{s2} - c_s)q_{s2}^* - \pi_{r2}^*$, the rise of $c_r$ and $\beta$ is first less than zero and then greater than zero. The color change represents the same meaning as Figure 5b.

Thus, when the production cost of new products is small enough, the optimal profits for wholesale gas and new products are greater than those for ordinary products, and the earnings of new products are greater than those of wholesale products. However, at this time, the producers and recyclers have optimal profits, and the market will reach a stable state (see Figures 5 and 6). Unlike reference [18], this point does not use the coordination mechanism to receive the profits.

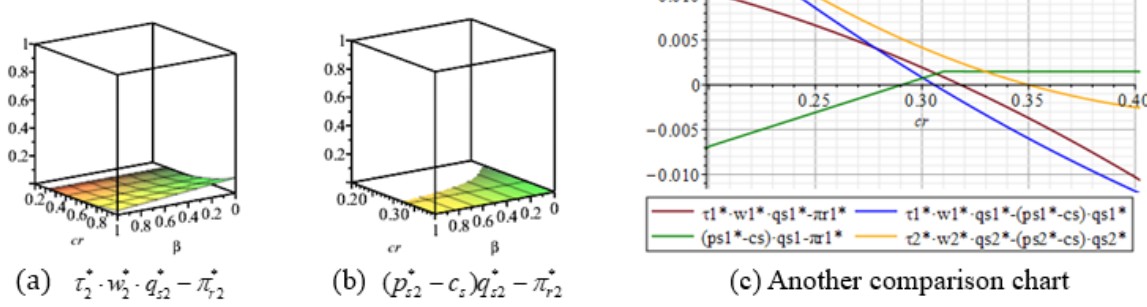

**Figure 6.** The internal profit comparison and analysis chart for $\pi_r$ and $\pi_m$.

### 7. Conclusions and Final Thoughts

*7.1. Conclusions*

This text explores the optimization challenge faced in the waste recycling process. It begins by analyzing the optimal price, sales volume, and profit achieved through two separate sales channels for new and ordinary products. The next step involves an examination of the influence of factors such as recovery ratio, costs, and others on the optimal results when considering or not considering carbon emissions. The final section provides a conclusion on the decision-making for the sales volume, price, and profit of both new and ordinary products through further analysis and examples. The conclusions are as follows:

(1) From the perspective of the proportion of recovered exhaust gas, the general proportion of recovered exhaust gas will directly affect the general sales volume and the optimal sales volume. Otherwise, it also indirectly serves as the intermediary of cost and policy when it is the optimal proportion of waste gas recovery. Among these, the cost of gas-containing metal elements has a positive impact on the optimal proportion of recovered exhaust gas, the cost of new products has a negative effect on the optimal proportion of recovered exhaust gas, and the impact of national policies that are not affected by other factors is a mandatory one.

(2) Common products and new products under optimal sales volume and optimal sales volume are affected by different reasons and degrees in product sales. Technical factors positively impact new products, while transaction factors hurt both. Specifically, the cost of gas-containing metal elements and the general cost of wholesale gas negatively affect the optimal sales volume of ordinary products. Additionally, the optimal sales volume of new products is positively affected by the proportion of carbon quota, the proportion of gas concentration (conversion), and the cost of gas-containing metal elements. Still, it is negatively affected by the production cost of new products. Besides, the proportion of the recovered waste gas positively affects the general sales volume of new products.

(3) The decision-making for recyclers, in terms of optimal prices and profits, is primarily driven by revenue. For example, when the production cost of new products rises, it is beneficial for the optimal price of ordinary products and the optimal profit of recyclers without considering the carbon quota. When the production cost of new products drops, the yield of new products is more significant than that for ordinary products, yet both producers and recyclers benefit.

(4) New and ordinary products are constrained by each other, resulting in mutual influence in the marketplace. Just as ordinary products sell poorly, new products sell well. Thus, new products will drive ordinary products out of the market without considering the carbon quota. On the contrary, the two will balance each other in the market when considering the carbon quota.

Overall, the cost of gas-containing metal elements positively impacts the optimal sales volume of ordinary products and new products as well as the optimal proportion of recovered exhaust gas. Moreover, the production cost of new products has a negative impact on all three aspects.

*7.2. Theoretical Implications*

This paper presents its theoretical significance through the lenses of supply chain, management, and enterprise. Firstly, it introduces a novel approach to studying the traditional supply chain by focusing on the gas supply chain. The analysis highlights that while most studies on waste recovery in the supply chain concentrate on government subsidies and technology inputs, the proportion of recovered waste gas offers a fresh perspective on the development of the gas supply chain and the primary business of the gas company.

Secondly, product quality is guaranteed by the gas concentration (conversion) ratio during the process of remanufacturing and recycling, which provides reference suggestions

for enterprises to grasp product quality better and ensure the benefits of technology. Unlike the conventional methods of defining remanufactured products with low recycling quality, this study shows how product quality can be improved and benefits guaranteed from the perspective of gas recovery and conversion.

Finally, based on the waste recycling and reuse model, the influence of various model factors on decision-making without considering the carbon quota and the carbon quota was studied and expanded the research content in the management field. Most of the existing literature uses policy as the influencing factor when exploring the influence on the model of efficiency and profit [41]. Compared to the previous studies, the current research pays more attention to the impact of specific factors in the research process than the impact of policies.

### 7.3. Managerial Implications

The study has several managerial implications. Firstly, the proportion of waste gas recovered is a new entry point for startups and spinoffs. The study sheds light on the changes brought about by new products entering the market, as well as providing strategies for managing potential risks by sub-dividing the cost of gas containing metal elements. This focus coincides with Yilmaz Bayar et al. and their use of the rate of recovery to deal with optimal decision-making for the recovery, treatment, and reuse of waste materials [22]. Companies can thus ensure product quality by starting with technology that recovers the proportion of the waste gas.

Secondly, the government can adjust its related policies according to actual needs to ensure the interests of all parties and achieve a tripartite environmental–social–economic ecosystem. This is particularly important in the current epidemic, where it is crucial to consider the positive balance of all stakeholders. For example, the Suzhou Municipal Government has taken the lead in supporting small and medium-sized enterprises through its "ten policy opinions" during the business risk period [64]. Producers and recyclers can also achieve a win-win situation, but the government must implement policies in a reasonable manner to attract more participants.

Lastly, recyclers play multiple roles, sharing the pressure for recycling and reusing waste. In this paper, the recycler recycles waste gas, acts as the dealer to sell new products, and even undertakes other manufacturers' work to refine gas. According to solid law, reliable waste treatment only needs to meet the relevant environmental protection requirements and national control standards [65], thus, the task undertaken by the recycler here is exceptionally significant for ecological sustainability.

### 7.4. Limitation of Research

This paper's limitations suggest one direction for future research. First of all, the model only studied a single period due to the limit of the recovery stage, and there is a lot of space still to search for two periods, multi-periods, and an infinite period. Secondly, more main bodies related to processing and transportation can be designed to play games, but only a producer and a recycler exist. Finally, the Stackelberg model was used to study this model, but other models have not yet had a thorough inquiry, i.e., the Nash equilibrium, and these could also offer significant additional research on the topic discussed here. Not only that, this paper is limited by the carbon quota, but some future changes in laws and policies will create a different environment for research, which is a point of entry.

**Author Contributions:** K.W. and X.L. were responsible for the research design and conception. H.L. made the bilingual revision of the paper. All authors have read and agreed to the published version of the manuscript.

**Funding:** This research received no external funding.

**Institutional Review Board Statement:** Not applicable.

**Informed Consent Statement:** Not applicable.

**Data Availability Statement:** Data available on request due to restrictions eg privacy or ethical.

**Conflicts of Interest:** The authors declare no conflict of interest.

## Appendix A

According to the Stackelberg model, the manufacturer is the leader, and the recycler is the follower. Therefore, the proportion of waste gas $\tau_1^*(\tau_2^*)$ recovered by the recycler must be determined first. Secondly, according to the reverse induction method, select the wholesale gas price of the manufacturer $w_1^*(w_2^*)$. Then, determine the number of ordinary products $q_{s1}^*(q_{s2}^*)$ sold by the manufacturer to the market. Finally, the formula $q_r = \beta \cdot \tau \cdot q_s$, $\beta \in (0, 1]$ determines the quantity $q_{r1}^*(q_{r2}^*)$. In addition, the price $p_{s1}^*(p_{s2}^*)$ at which the manufacturer sells ordinary products to the market and the price $p_{r1}^*(p_{r2}^*)$ at which the recycler sells new products.

*Appendix A.1. Without Considering the Carbon Quota*

Substituting Formula (2) into Formula (6) to obtain:

$$\pi_{r1} = \theta[1 - q_{s1}(\tau_1 + 1)] \cdot q_{s1} \cdot \tau_1 - c_r \cdot \tau_1 \cdot q_{s1} - w_1 \cdot \tau_1 \cdot q_{s1} \tag{A1}$$

$$\frac{\partial \pi_{r1}}{\partial \tau} = -\theta \cdot \tau_1 \cdot q_{s1}{}^2 + \theta \cdot [1 - q_{s1} \cdot (\tau_1 + 1)]q_{s1} - c_r \cdot q_{s1} - c_t \cdot q_{s1} - w_1 \cdot q_{s1} \tag{A2}$$

If $0 < \theta < 1$, $q_s > 0$, it can be seen that $\frac{\partial^2 \pi_{r1}}{\partial \tau_1^2} = -2q_{s1}^2 \cdot \theta < 0$, $\theta < 0$, so there is an optimal solution to the profit function of the recycler:

$$-\theta \cdot \tau_1 \cdot q_{s1}^2 + \theta \cdot [1 - q_{s1}(\tau_1 + 1)]q_{s1} - c_r \cdot q_{s1} - c_t \cdot q_{s1} - w_1 \cdot q_{s1} = 0 \tag{A3}$$

Can be obtained by item transfer:

$$q_{s1} = -\frac{c_t + c_r - \theta + w}{\theta(2\tau + 1)} \tag{A4}$$

According to the inverse induction method, replace Formula (3) with Formula (4) to obtain:

$$\pi_{m1} = -\frac{w_1(q_{s1} \cdot \theta + c_r + \delta \cdot c_t - \theta + w_1)}{2\theta} + [1 - q_{s1}(-\frac{q_{s1} \cdot \theta + c_r + \delta \cdot c_t - \theta + w_1}{2q_{s1}} + 1) - c_s]q_{s1} \tag{A5}$$

From Equation (A1), we can obtain that the Hessian matrix of the profit function of the manufacturer is

$$\begin{bmatrix} -\frac{1}{\theta} & 0 \\ 0 & \theta - 2 \end{bmatrix} \tag{A6}$$

If $D_1 = -\frac{1}{\theta} < 0$, $D_2 = -\frac{1}{\theta}(\theta - 2) > 0$, the Hessian matrix is negative definite, and the profit function of the manufacturer is concave, so there is an optimal solution:

$$-\frac{q_{s1} \cdot \theta + c_r + \delta \cdot c_t - \theta + w_1}{2\theta} - \frac{w}{2\theta} + \frac{q_{s1}}{2} = 0 \tag{A7}$$

$$-\frac{w_1}{2} + [\frac{q_{s1} \cdot \theta + \delta \cdot c_r - \theta + w_1}{2q_{s1}} - 1 - q_{s1}(-\frac{\theta}{2q_{s1}} + \frac{q_{s1} \cdot \theta + c_r + \delta \cdot c_t - \theta + w_1}{2q_{s1}^2})]q_{s1} + 1 - q_{s1}(-\frac{q_{s1} \cdot \theta + c_r + \delta \cdot c_t - \theta + w_1}{2q_{s1}^2} + 1) - c_s = 0 \tag{A8}$$

Solve Equations (A3), (A7) and (A8) to obtain:

$$\tau_1^* = \frac{[(\delta - 1)\theta - 4\delta + 2]c_t + [(1 - \delta)\theta + 2\delta - 4]c_r + 2\theta c_s}{2\theta(\delta c_t + c_r - 2c_s - \theta + 2)} \tag{A9}$$

$$w_1^* = -\frac{\delta c_r}{2} - \frac{c_t}{2} + \frac{\theta}{2} \tag{A10}$$

$$q_{s1}^* = -\frac{c_r - 2c_s + \delta c_t - \theta + 2}{2(\theta - 2)} \tag{A11}$$

Therefore, according to Formula (1):

$$q_{r1}^* = \frac{[(1-\delta)\theta + 4\delta - 2]c_t + [(\delta-1)\theta - 2\delta + 4]c_r - 2c_s}{4\theta(\theta - 2)} \tag{A12}$$

Corollary 1 proved: $w_1^*$ is a monotonic subtractive function of $c_t$.

Corollary 2 proved: $q_{s1}^*$ is the monotonic decreasing function of $c_t$, and $q_{r1}^*$ is the monotonic increasing function of $c_t$.

Corollary 3 proved: $q_{s1}$ is a monotonic subtractive function of $w_1$.

Corollary 4 proved: From the value of $w_1^*$, $-\delta c_r - c_t + \theta > 0$. When $\delta > 0$, $c_r < \frac{\theta - c_t}{\delta}$, When $\delta < 0$, $c_r > \frac{\theta - c_t}{\delta}$. Similarly, from the value of $q_{s1}^*$, $c_r > 2c_s - \delta \cdot c_t + \theta - 2$. Because $2c_s - \delta \cdot c_t + \theta - 2$ and $(\frac{\theta - c_t}{\delta})$, one of them may be greater than zero or less than zero.

So, Proposition 1 and Proposition 2 are proven.

Corollary 5 proved: $\tau_1^*$ is the monotonic increasing function of $c_t$.

Corollary 6 proved: $q_{s1}$ is a monotonic subtractive function of $\tau_1$, and Formula (1) $q_{r1} = \tau_1 \cdot q_{s1}$ shows that $q_{r1}$ is the monotonic increasing function of $\tau_1$.

So, Proposition 3 is proven.

Substituting Equations (A9) and (A11) into Formulas (2) and (3) to obtain:

$$p_{s1}^* = \frac{(\delta - 1)c_t + (1 - \delta)c_r + 2(c_s + 1)}{4} \tag{A13}$$

$$p_{r1}^* = \frac{[(3\delta - 1)\theta - 4\delta + 2]c_t + [(3 - \delta)\theta + 2\delta - 4]c_r - 2\theta c_s + 2\theta^2 - 4\theta}{4\theta - 8} \tag{A14}$$

Corollary 7 proved: $p_{s1}^*$ combined with (A11), it can be seen that $p_{s1}^*$ and $q_{s1}^*$ are monotonically increasing functions of $c_s$. From the combination of $p_{r1}^*$ and (A12), it can be seen that $p_{r1}^*$ and $q_{r1}^*$ are monotonic subtractive functions of $c_s$.

So, Proposition 4 is proven.

Substituting Equations (A9)–(A11) into Equations (A1) and (A6):

$$\pi_{m1}^* = \frac{1}{\delta\theta(\theta-2)}\left\{(2-4c_s)\theta^2 + [-(c_r-c_t)^2\delta^2 + (2c_r^2 - 4c_r \cdot c_t + 4c_s \cdot c_t + 2c_t^2)\delta - c_r^2 + (4c_s + 2c_t)c_r - c_t^2 - 4(c_r - c_t)^2]\theta + 2[(c_r - 2c_t)\theta - 2c_r + c_t](c_r\theta + c_t)\right\} \tag{A15}$$

$$\pi_{r1}^* = \frac{\{\theta[(c_r - c_t)\delta - c_r - 2c_s + c_t] + (4c_t - 2c_r)\delta + 4c_r - 2c_t\}^2}{16\theta(\theta - 2)^2} \tag{A16}$$

*Appendix A.2. Considering the Carbon Quota*

Substituting Formula (2) into Formula (7) gives:

$$\pi_{r2} = -\beta \cdot c_t \cdot \tau_2 \cdot \delta \cdot q_{s2} + \theta[1 - q_{s2}(\tau_2 + 1)]\beta \cdot q_{s2} \cdot \tau_2 - \beta \cdot c_r \cdot \tau_2 \cdot q_{s2} - \beta \cdot w_2 \cdot \tau_2 \cdot q_{s2} \tag{A17}$$

$$\frac{\partial \pi_{r2}}{\partial \tau} = -\beta \cdot c_t \cdot \delta \cdot q_{s2} - \theta \cdot \beta \cdot \tau_2 \cdot q_{s2}^2 + \theta[1 - q_{s2}(\tau_2 + 1)]\beta \cdot q_{s2} - \beta \cdot c_r \cdot q_{s2} - \beta \cdot w_2 \cdot q_{s2} \tag{A18}$$

If $0 < \theta < 1$, and $q_{s2} > 0$, $\frac{\partial^2 \pi_{r2}}{\partial \tau^2} = -2 \cdot \beta \cdot q_{s2}^2 \cdot \theta < 0$. Therefore, there is an optimal solution to the profit function of the recycler

$$-\beta \cdot c_t \cdot \delta \cdot q_{s2} - \theta \cdot \beta \cdot \tau_2 \cdot q_{s2}^2 + \theta[1 - q_{s2}(\tau_2 + 1)]\beta \cdot q_{s2} - \beta \cdot c_r \cdot q_{s2} - \beta \cdot w_2 \cdot q_{s2} = 0 \tag{A19}$$

According to the inverse induction method, we can also have Equations (A5), (A6), (A7) and (A8) to solve Equations (A7), (A8) and (A19), for obtaining:

$$\tau_2{}^* = \frac{\theta c_s - \delta c_t - c_r}{\theta(\delta c_t + c_r - 2c_s - \theta + 2)} \tag{A20}$$

$$w_2{}^* = -\frac{\delta c_r}{2} - \frac{c_t}{2} + \frac{\theta}{2} \tag{A21}$$

$$q_{s2}{}^* = -\frac{c_r - 2c_s + \delta c_t - \theta + 2}{2(\theta - 2)} \tag{A22}$$

Therefore, according to Formula (1):

$$q_{r2}{}^* = \frac{\beta(-\theta c_s + \delta c_t + c_r)}{4\theta(\theta - 2)} \tag{A23}$$

Substituting Equations (A20) and (A22) into Formulas (2) and (3) to obtain:

$$p_{s2}{}^* = \frac{c_s + 1}{2} \tag{A24}$$

$$p_{r2}{}^* = \frac{\theta^2 + \theta(\delta \cdot c_t + c_r - c_s - 2) - \delta \cdot c_t - c_r}{2\theta - 4} \tag{A25}$$

Substituting Equations (A20)–(A22) into Equations (A17) and (A5):

$$\pi_{m2}{}^* = \frac{\theta^2(1 - 2c_s) + \theta[-2 - 2c_s{}^* + (2\delta \cdot c_t + 2c_r + 4)c_s] - (\delta \cdot c_t + c_r)^2}{4\theta(\theta - 2)} \tag{A26}$$

$$\pi_{r2}{}^* = \frac{\beta(-\theta \cdot c_s + \delta \cdot c_t + c_r)^2}{4\theta(\theta - 2)^2} \tag{A27}$$

Corollary 8 proved: (1) From the values of $w_1{}^*$, $w_2{}^*$ and $q_{s1}{}^*$, $q_{s2}{}^*$. (2) Obtained in the same way as corollary 1, corollary 2, corollary 3, corollary 6, and corollary 7, which is omitted.

Corollary 9 proved: (1) $p_{s1}{}^* - p_{s2}{}^* = p_{r1}{}^* - p_{r2}{}^* = -\frac{(1-\delta)(c_r - c_t)}{4}$; (2) $\pi_{m1}{}^* - \pi_{m2}{}^* = -\frac{(\delta - 1)^2(c_r - c_t)^2}{4} < 0$.

Corollary 10 proved: From the values of $\tau_2{}^*$, $c_t(c_r)$ is monotonically increasing function of $\tau_2{}^*$.

So, Proposition 5 is proven.

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
