# Peer review of "Research on Supply Chain Decisions for Production Waste Recovery and Reuse Based on a Recycler Focus"

_sustainability, doi:10.3390/su15043320_

Round 1

Reviewer 1 Report

The abstract does not correspond to the article.

The introduction needs to be shortened.

The literature review is long and boring.

Fig. 1 is on page 6, but on page 12 is again fig.1.

On the basis of which data was Fig. 4 constructed?

Reviewer 2 Report

This study investigated a supply chain decisions for production waste recovery and reuse. The topic is very current and well processed.

Four main views were explored (lines 92-101). They were formulated and set correctly.

The aim, methods, results and conclusions of the paper are correct.

I recommend adding a literature review focused on „waste audit in supply chain“

I recommend adding  a subchapter "limitation of research".

Reviewer 3 Report

The paper  provides a new concept for waste recovery and recycling taking metal recovery from gas waste as an example.

The paper is well structured and the methods are well described.

However, the abstract is not well written please reformulate the abstract.

Authors have used production waste recovery through the whole manuscript which is not correct. waste recovery  or products recovered from waste are more adequate.

figure presented in page 12 should be figure 3 and not figure 1

Round 2

Reviewer 1 Report

The authors responded to all comments and incorporated them into the article.